# Effects of Autologous Microfragmented Adipose Tissue on Healing of Tibial Plateau Levelling Osteotomies in Dogs: A Prospective Clinical Trial

**DOI:** 10.3390/ani13132084

**Published:** 2023-06-23

**Authors:** Luca Pennasilico, Caterina Di Bella, Sara Sassaroli, Alberto Salvaggio, Francesco Roggiolani, Angela Palumbo Piccionello

**Affiliations:** 1School of Bioscience and Veterinary Medicine, University of Camerino, 62024 Matelica, Italy; luca.pennasilico@unicam.it (L.P.); caterina.dibella@unicam.it (C.D.B.); sara.sassaroli@unicam.it (S.S.); 2TéchneVet, 62024 Matelica, Italy; alberto.salvaggio@icloud.com (A.S.); roggiolanifrancesco@gmail.com (F.R.)

**Keywords:** microfragmented adipose tissue, regenerative medicine, bone healing, osteotomy, cranial cruciate ligament rupture, dog

## Abstract

**Simple Summary:**

Bone healing is a proliferative physiological process mediated by a complex interaction between biological and biomechanical mechanisms. Major complications of fracture repair and osteotomies include delayed union, non-union, and malunion. Adipose stem cells (ADSCs) have been shown to have positive effects on bone healing. In response to microenvironment signals, ADSCs can differentiate into osteogenic-like cells; moreover, they produce several cytokines and growth factors that are directly involved in bone healing. In clinical practice, ADSCs are often not administered as a pure isolate but rather as a constituent of microfragmented adipose tissue (MFAT) due to processing time and legislative restrictions. The purpose of this prospective, randomised, blinded, in vivo clinical study was to estimate the effect of autologous MFAT on bone healing in dogs who underwent tibial plateau levelling osteotomy. The results suggest that administration of MFAT is safe, cost-effective, minimally invasive, saves time, and could accelerate the bone healing of osteotomies or acute fractures. Additionally, the canine model adopted in this study could play a key role in developing successful treatments for translational medicine.

**Abstract:**

The aim of this study was to evaluate the effects of autologous microfragmented adipose tissue (MFAT) applied after mechanical fragmentation and assess these effects radiographically in bone healing in dogs subjected to tibial plateau levelling osteotomy (TPLO). Twenty dogs with unilateral cranial cruciate ligament disease were enrolled and randomly assigned to the treatment group (MFAT) or the control group (NT). The MFAT group underwent TPLO and autologous MFAT intra-articular administration, while the NT group underwent TPLO alone. Adipose tissue was collected from the thigh region, and MFAT was obtained by mechanical fragmentation at the end of the surgery. The patients were subjected to X-ray examination preoperatively, immediately postoperatively (T0), and at 4 (T1) and 8 (T2) weeks postoperatively. Two radiographic scores that had previously been described for the evaluation of bone healing after TPLO were used. A 12-point scoring system (from 0 = no healing to 12 = complete remodelling) was used at T0, T1, and T2, while a 5-point scoring system (from 0 = no healing to 4 = 76–100% of healing) was used at T1 and T2. The median healing scores were significantly higher at T1 and T2 for the MFAT group compared with the NT group for the 12-point (*p* < 0.05) and 5-point (*p* < 0.05) scoring systems. The intra-articular injection of autologous microfragmented adipose tissue can accelerate bone healing after TPLO without complications.

## 1. Introduction

Cranial cruciate ligament rupture (CCLR) is the most common acquired cause of pelvic limb lameness in dogs [1,2]. The aetiology is not entirely clear, but degenerative or conformational causes (such as excessive inclination of the tibial plateau, intercondylar notch stenosis, or insufficient nutritive supply core of the ligament) are most commonly associated with a traumatic event [3,4,5,6]. Since 1952, several surgeries have been described for the treatment of CCLR in dogs; they can be subdivided into intra- or extracapsular reconstruction of the ligament and corrective osteotomies [7,8,9,10]. The latter are the most recent conception: they aim to modify the stifle joint biomechanics by neutralising the cranial tibial thrust (CTT) [11,12]. Among these procedures, tibial plateau levelling osteotomy (TPLO) is one of the most used and shows satisfactory outcomes [13,14,15,16].

Although it is a standardised surgical procedure with a high clinical success rate, TPLO is not without its complications; reported among these are mechanical failure with a loss of reduction prior to the attainment of bone union and delayed bone union or non-union [17,18,19,20]. Strategies to treat these consolidation delays are continuously being evaluated [19,21]. Since TPLO is executed following a standardised protocol, this standardisation of the technique facilitates the study of potential promotors of bone healing in a clinical setting [19,22,23,24,25].

Numerous studies have highlighted the osteogenic power of adipose stem cells (ADSCs) and their use, alone or in association with biomaterials, in the repair of bone defects or to increase the speed of bone healing [26,27,28]. Based on in vivo and in vitro studies, the induction of ADSCs leads to several changes: increased expression of osteogenic genes, including runt-related transcription factor 2 (RUNX2) and osterix (OSX); increased alkaline phosphatase activity and calcium phosphate mineralised extracellular matrix; and increased osteocalcin, osteopontin, and collagen type I protein [29,30]. In general, mesenchymal stem cells have a fundamental role in primary and secondary bone healing, differentiating into osteoblasts and chondrocytes to complete the repair of bone [31,32,33].

In clinical practice, autologous ADSCs are often not administered as a pure isolate but rather as a constituent of the stromal vascular fraction (SVF) [34,35]. The advantages of SVF include its abundant stem cell content, ease of extraction, the availability of tissue, and the minimal invasiveness of the harvest [36,37]. Certain articles have reported promising results regarding the application of SVF in bone healing [38,39]. Saxer et al. [39] demonstrated that SVF cells, without expansion or exogenous priming, can spontaneously form bone tissue and vessel structures within a fracture microenvironment. In addition, Kim et al. [40] highlighted the osteogenic potential of SVF in vitro and the increase in bone healing with a scaffold seeded with SVF compared with the scaffold alone in a rabbit model.

SVF isolation involves the use of enzymatic digestion of adipose tissue, a process that is expensive, time-consuming, and strictly regulated by the good manufacturing procedure of the European Parliament (EC regulation no. 1394/2007). Therefore, it can only be performed in specialised laboratories [41,42]. In this scenario, companies have developed automatic closed devices that allow the mechanical processing of adipose tissue [43,44]. The mechanical disaggregation of fat tissue results in microfragmented adipose tissue (MFAT), sometimes referred to as SVF from MFAT [37,44]. Microfragmented adipose tissue contains all the cells found in adipose tissue, including adipocytes, and preserves the microenvironment of the perivascular niche. For this reason, it has a greater paracrine and immunomodulatory effect then enzymatic SVF, albeit with fewer stem cells [44,45,46,47].

In human medicine, studies have shown the potentiality of MFAT administration in plastic surgery, wound and fistula healing, and in management of degenerative chronic diseases (osteoarthrosis and diabetes) [48,49,50,51,52]. On the other hand, the scientific literature concerning the use of MFAT in veterinary medicine is very limited [53,54,55,56]. To our knowledge, there have been no previous studies evaluating the effect of MFAT on bone healing in either human or veterinary medicine. The aim of this study was to evaluate the effects of autologous adipose micrografts, obtained by mechanical fragmentation, on the healing of procedural osteotomies in dogs subjected to TPLO.

## 2. Materials and Methods

### 2.1. Ethics Statement

This clinical study was approved by the Ethics Committee for Clinical Study in Animal Patients of the University of Camerino with approval number Prot. 06/23. Informed consent was obtained from the owners of all the dogs.

### 2.2. Animals

Twenty mixed-breed dogs with unilateral naturally occurring CCL disease were prospectively enrolled in the study. The diagnosis of complete CCL rupture was based on orthopaedic examination, and it was confirmed intraoperatively. Each dog was subjected to general, orthopaedic, and neurological physical examination, in order to exclude other relatable pathologies. In addition, the blood cell count and biochemistry profile were obtained. Dogs with concurrent orthopaedic, neurological, or metabolic disease were excluded from the study, as well as dogs with contralateral CCL disease. Only dogs with body condition scores between 4 and 6 (on a scale of 1–9) were considered. Ten dogs were randomly assigned to the treatment group (MFAT) and the other ten to the control group (NT).

### 2.3. Surgical Procedures

Each TPLO was performed by the same expert orthopaedic surgeon as described previously [13].

The subjects were positioned in dorsolateral recumbency. The affected limb and surgical field were aseptically prepared. A craniomedial surgical approach to the proximal tibia was performed with sartorius muscle dissection and popliteus muscle detachment from the bone caudal aspect. Cefazoline (Teva s.r.l., Milan, Italy) was administered intravenously (22 mg/kg) approximately 30 min before the skin incision, and a second administration was given after 90 min.

The plate and radial saw blade were selected by the surgeon according to the reported surgical technique indications and the surgeon’s choice [57]. DePuy Synthes TPLO Locking Plates (DePuy Synthes, Oberdorf, Switzerland) were used. All osteotomies were compressed with the placement of two conventional screws in compression. During the osteotomy and drilling, copious saline solution irrigation was used to prevent thermal bone necrosis. The surgical site was then sutured routinely using USP 2/0 absorbable monofilament thread (polydioxanone). Arthrotomy or arthroscopy was not performed on any dog. At the end of surgery, a soft padded bandage was applied for 48 h in order to reduce postoperative oedema. Rescue analgesia was administrated if needed. In the postoperative period, the dogs received antibiotic therapy for 6 days (Cefadroxil 20 mg/kg BID OS, Cefa-Cure Tabs, MSD Animal Health, Segrate, Italy) and non-steroidal anti-inflammatory therapy for 10 days (carprofen 3 mg/kg SID OS; Rymadil, Zoetis s.r.l., Roma, Italy).

### 2.4. Isolation, Preparation and Inoculation of MFAT

During TPLO, about 4 g of thigh subcutaneous fat was harvested from dogs of the MFAT group. The fat was processed with the Rigenera^®^ (Human Brain Wave, Turin, Italy) system, a mechanical disruptor of biological tissue. It consists of a motorised apparatus that allows a sterile and disposable capsule (Rigeneracons^®^) to mechanically disrupt the tissue placed inside. Each Rigeneracons^®^ is made up of a helical blade controlled by an electric motor that rotates it at 80 rpm, thus allowing a precise, uniform, and constant cut. Furthermore, at the end of each helix there is a metal filter containing 100 holes of about 50 µm, each of which has six micro-scalpels. The desegregated and filtered tissue is collected at the bottom of the capsule and, through a syringe connector, it is possible to aspirate and use the preparation. Approximately 1 mL of the obtained MFAT was inoculated intra-articularly in dogs of the MFAT group; the dogs of the NT group did not receive an inoculation. Intra-articular inoculation of the stifle joint was performed with a medial para-patellar approach and an 18G needle.

### 2.5. Radiographic Evaluations

The patients were subject to an X-ray exam preoperatively, immediately after surgery (T0), and at 4 (T1) and 8 (T2) weeks after surgery. Radiographs of the stifle joint were taken in the mediolateral and caudocranial views with the animal sedated. Only patients that showed good compression (no visible gap) on a postoperative X-ray were included in the study. The mediolateral view was performed with the stifle and tarsus joints flexed at 90° and the femoral condyles superimposed over each other. The caudocranial view involved the medial cortex of the calcaneus intersecting the middle of the tibial trochlear with the patella superimposed centrally between the femoral condyles. At T0, each patient was evaluated for the degree of osteoarthritis (OA) according to a modified Kellgren–Lawrence scale from 0 (absence of OA) to 4 (highest degree of OA) [58].

Two radiographic scores, described previously [19,22] for the evaluation of bone healing after TPLO, were adopted by an expert (>20 years of experience) blinded radiologist. A 12-point scoring system (from 0 = no healing to 12 = complete remodelling) was used at T0, T1, and T2 (Table 1), while a 5-point scoring system (from 0 = no healing to 4 = 76–100% healing) was used at T1 and T2 (Table 2).

### 2.6. Statistical Analysis

A sample size analysis was performed considering our preliminary data in dogs. Power calculation was conducted for a two-tailed *t*-test considering the mean of the 5-point scoring system as the reference parameter, with a power of 0.95 and an alpha error of 0.05 (G*Power Version 3.1.9.3). The test suggested that a minimum of 16 dogs could be sufficient to detected significant differences, with an effect size of 1.30 [59]. Statistical analysis was performed using MedCalc software version 9.2.10. All data were tested for normality with the Shapiro–Wilk test and are reported as the mean ± standard deviation. Parametric data were analysed with a two-way analysis of variance (ANOVA) for repeated measurements. A *p*-value < 0.05 was considered to be statistically significant.

## 3. Results

### 3.1. Enrolled Patients

The following patients were enrolled: four American Staffordshire Terriers, two American Pit Bull Terriers, two German Shepherds, three Labrador Retrievers, one Golden Retriever, one Rottweiler, two Cane Corso Italiano, two Deutscher Boxers, and three mixed-breed dogs. The mean weight was 30.37 ± 8.25 kg (31.98 ± 7.38 kg for the MFAT group; 28.75 ± 9.13 kg for the NT group; *p* = 0.396), and the mean age was 4 ± 1.74 years (3.9 ± 2.2 year for the MFAT group; 4.1 ± 1.52 years for the NT group; *p* = 0.806) at the time of surgery.

### 3.2. Surgical Variables

The mean preoperative tibial plateau angle (TPA) was 27° ± 2.06° (range 23–30°) and the mean postoperative TPA was 4.8° ± 1.4° (range 2–7°). There were no differences in the preoperative and postoperative TPA between the MFAT and NT groups (*p* = 0.92 and *p* = 0.789, respectively). Eighteen 3.5-mm TPLO plates (six 3.5-mm broad plates, eight 3.5-mm standard plates, and four 3.5-mm small plates) and two 2.7-mm TPLO plates were used. The mean surgical time was 70 ± 2.31 min in the MFAT group and 64 ± 3.97 in the NT group. The average preparation time for MFAT was 5 min (range 4–6 min).

### 3.3. Radiographic Examination

No subjects were excluded; all subjects showed satisfying radiographic compression of the tibial osteotomy immediately after surgery (T0). Preoperatively there were no differences in the osteoarthritis degree between the MFAT and NT groups (1.9 ± 1.1 and 2 ± 1.05, respectively) (*p* = 0.838). Radiographic evaluation was also performed at 4 and 8 weeks after surgery (T1 and T2, respectively) for all patients. Bone healing was significantly (*p* < 0.05) increased at T1 and T2 compared with T0 in both the MFAT and NT groups (Figure 1 and Figure 2). At T1, the 12- and 5-point scores were significantly (*p* < 0.05) higher for the MFAT group (7.28 ± 0.75 and 3 ± 0.57, respectively) compared with the NT group (5.71 ± 1.38 and 2.14 ± 0.69, respectively). At T2, the MFAT group showed a greater bone healing based on higher 12- and 5-point scores (10.85 ± 1.77 and 3.85 ± 0.37, respectively) than the NT group (9.71 ± 1.6 and 3.28 ± 0.48, respectively) (*p* < 0.05) (Figure 3 and Figure 4).

There were significant differences (*p* < 0.05) in the subjective degree of callus formation and the degree of rounding of the distal step at the osteotomy site between the MFAT group (2.6 ± 0.51 and 1.3 ± 0.48, respectively) and the NT group (1.6 ± 0.69 and 1 ± 0.66, respectively) at T1. At T2, the differences in the subjective degree of callus formation and the degree of rounding of the distal step at the osteotomy site were significantly different (*p* < 0.05) between the MFAT group (3.6 ± 0.51 and 1.8 ± 0.42, respectively) and the NT group (2.9 ± 0.73 and 1.6 ± 0.51, respectively) (Figure 5 and Figure 6). In addition, there were significant differences (*p* < 0.05) in the 12-point score, the 5-point score, the subjective degree of callus formation, and the degree of rounding of the distal step at the osteotomy site in the same group between T1 and T2.

## 4. Discussion

Although some studies have demonstrated that ADSCs can promote bone healing in certain bone segments, this is the first study to evaluate the efficacy of an adipose autologous micrograft on the acute bone healing of osteotomy gaps in subjects undergoing TPLO. Our results show a significantly stronger osteogenic response at 30 days post-operatively compared with the group not having received MFAT. These results can be attributed to the effectiveness of the complex heterogeneous mixture of cells in MFAT: ADSCs, endothelial progenitor cells, smooth muscle cells, pericytes, fibroblasts, preadipocytes, myeloid cells, haematopoietic cells, monocytes, lymphocytes, and granulocytes, as described previously [60,61].

The stem cells contained in MFAT are directly involved in the bone remodelling process. Specifically, they can differentiate into osteoblasts to promote bone formation and to secrete various cytokines which stimulate the differentiation of progenitor cells into endothelial cells to promote neoangiogenesis, with a beneficial effect on bone healing. In particular, ADSCs upregulate the expression of angiogenic (hepatocyte growth factor [HGF] and vascular endothelial growth factor [VEGF]), haematopoietic (granulocyte colony-stimulating factor [G-CSF]), and bone-formation-promoting (bone morphogenetic protein 2 [BMP-2] and transforming growth factor beta [TGF-β]) cytokines [62]. In other words, ADSCs favour the deposition of bone tissue by stimulating osteogenesis and angiogenesis. The literature mostly describes the use of pure stem cells together with autologous or synthetic scaffolds. The results are promising, especially in the treatment of bone defects; however, Topleau et al. [63] recently reported that SVF can also induce osteogenesis without invasive techniques involving bone grafting or alloplastic scaffolds. Saxer et al. [39] have even demonstrated that SVF can form vascular structures and bone tissue within a fracture, without expansion or exogenous priming. MFAT increases TGF-β1 expression in rats with fractures and large bone defects. TGF-β1 is a growth factor that modulates bone healing primarily through the stimulation of undifferentiated mesenchymal cells by inducing osteoblast proliferation [64,65,66].

In our hands, the harvesting and preparation of the autologous subcutaneous tissue by mechanical disaggregation was simple and fast: it took no more than 6 min in total. Compared with bone marrow–derived mesenchymal stem cells, MFAT is an easily accessible and rich source of stem cells that can be prepared intraoperatively with minimal manipulation and no expansion. Overall, there are several characteristics of adipose tissue that have made it the most promising and appreciated source of stem cells: less painful sampling and a stem cell yield 500 times higher than that obtained from bone marrow (only 0.001–0.01% of the harvested bone marrow cells are mesenchymal stem cells), especially in elderly subjects [66,67,68,69,70]. For these reasons, in clinical routine, MFAT is often used rather than pure stem cells from bone marrow [45,71].

Several techniques, alone or in combination, have been described for nonenzymatic isolation of SVF, including filtration, centrifugation, vibration, disruption, shearing, and vortexing. Using a mechanical disruptor to obtain MFAT is easy, affordable (with a relatively low cost of equipment and consumables), and relatively quick [72]. Mechanical procedures for isolating SVF from adipose tissue represent a valid alternative to the enzymatic method, bypassing the ‘minimal manipulation’ limits strictly regulated by European laws [73]. In accordance with the human literature [74], in a recent study involving dogs [43], researchers sampled adipose tissue from the thigh. Subcutaneous adipose tissue is a more accessible and abundant source of stem cells than visceral adipose tissue [75]. Conversely, it represents a less suitable sampling site in older or emaciated subjects [76,77]. In our case, there were no difficulties in fat tissue collection, which was performed at the same time as the TPLO surgical approach.

Fat-tissue excision could provide a much higher number of nucleated cells compared with liposuction [78]. Liposuction involves the use of vacuum pressure that is more traumatic for the tissue, leading to adipocyte structural disruption [79]. On the contrary, direct excision represents a more delicate harvesting technique and creates large fat particles that preserve stromal constituents, providing structural support for adipocytes and proliferating stem cells [80]. In addition, a tumescent solution used for liposuction can reduce ADSC survival [81]. The fragmentation of adipose tissue represents a suitable mechanical approach with a cell yield and viability that are comparable to enzymatic procedures. According to the literature, we think that, to date, MFAT is the way to go in clinical regenerative applications involving adipose tissue due to processing time and legislative restrictions [82].

MFAT was inoculated intra-articularly immediately after surgery. As reported by Taroni et al. [25], after TPLO surgery, mesenchymal stem cells and their secreted substances contained in the synovial fluid come into contact with the osteotomy site. This route of administration is minimally invasive (only one injection with an 18G needle) and does not intervene directly at the osteotomy site. Therefore, this simple procedure lends itself to being used easily and speedily in a clinical setting.

In our study, the majority of subjects in both groups showed >70% radiographic bone healing with at least three continuous cortices 8 weeks after TPLO. However, the MFAT group showed increased bone healing and callus density compared with the NT group at both T1 and T2. In particular, the MFAT group showed an acceleration of bone healing at the first follow-up (4 weeks after TPLO). Callus formation, a ‘step’ distal to the osteotomy, and reduced visibility of the osteotomy line were significantly increased in the MFAT group at the latest follow-up (T2). The T1 results agree with a similar study evaluating the effect of ADSCs on bone healing in dogs undergoing tibial tuberosity advancement surgery; however, the results did not show significant differences between the treated group and the control group in the subsequent follow-up (at 60 days) [83]. Franco et al. [84] reported that the use of minimally invasive plate osteosynthesis (MIPO) in association with ADSC administration stimulated bone healing after a tibial fracture. They reported promising results: in their study, clinical bone healing occurred on average 28.5 and 70.3 days after treatment, respectively, in the stem cell and control groups. The use of a minimally invasive surgical approach in fracture repair preserves soft tissue blood supply (especially from muscle structures), periosteal vasculature, and early fracture haematoma. The MIPO technique and ADSC administration play a key role in promoting and accelerating callus formation, maturation, and remodelling by protecting the vascular network and enhancing neoangiogenesis. These two techniques in combination could certainly yield very interesting results and, above all, fully embrace the goal of minimal invasiveness if MFAT is used as source of ADSCs.

In contrast to some previous studies [23,85], which evaluated radiographic bone healing after TPLO using nonspecific scales, we assessed radiographic assessment by using two radiographic scores that have been described for the evaluation of bone healing after TPLO with more accurate results and a minimal margin of error. For this reason, the radiographic examination included two orthogonal TPLO views (mediolateral and caudocranial). We chose to assess bone healing with X-ray examination because it is widely available and has great practicality.

Among the limitations of the study, we can include the lack of a second-level diagnostic technique such as computed tomography and a histology that could characterise the formation of the bone callus. However, according to Italian legislation, clinical patients cannot be subjected to radiation or a bone biopsy if they do not need it (i.e., if it is only for research purposes). Other limitations of this study include the small number of cases enrolled and the lack of an intermediate time between T1 and T2.

In the future, it might be interesting to evaluate the progression of osteoarthritis in the knee joint treated with MFAT after TPLO surgery compared with a control group.

## 5. Conclusions

In conclusion, we can state that the mechanical disaggregation of adipose tissue represents a rapid and easy-to-perform method to prepare autologous MFAT that accelerates bone healing after an osteotomy without complications. Additional studies are needed to truly understand the effect of MFAT on bone healing in pathological conditions that can lead to delayed union or non-union.

## Figures and Tables

**Figure 1 animals-13-02084-f001:**
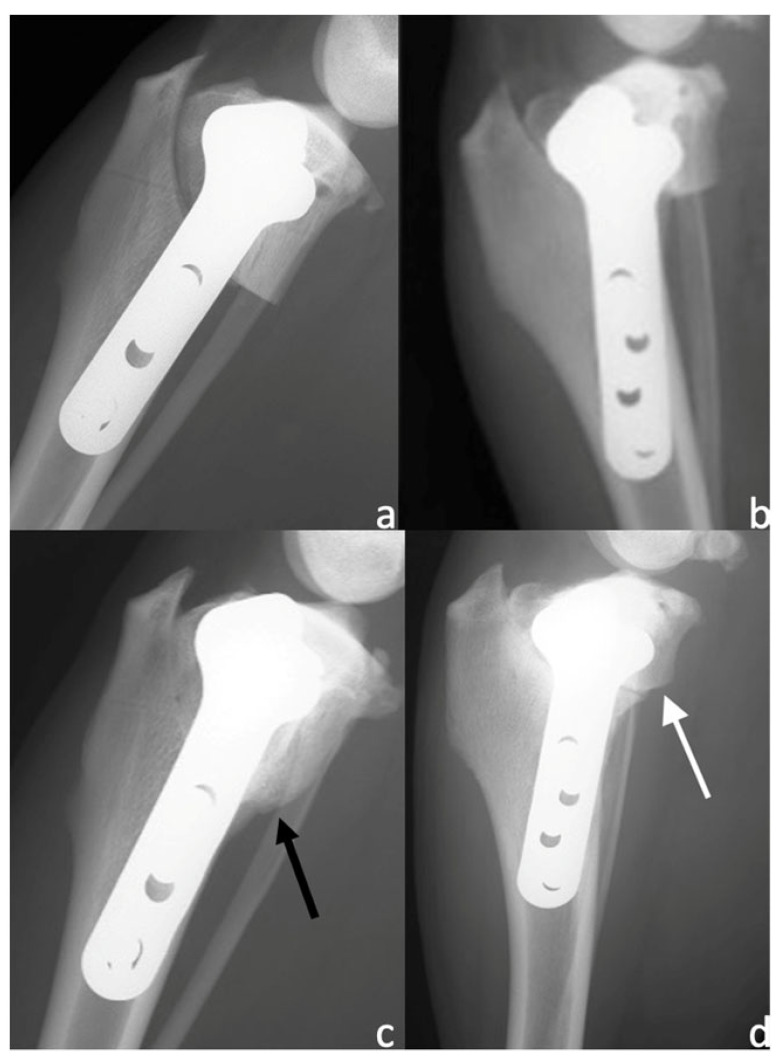
Mediolateral view of subjects treated with microfragmented adipose tissue (MFAT) immediately after tibial plateau levelling osteotomy (TPLO) (**a**) and after 8 weeks (**c**), showing a significant degree of rounding of the distal step at the osteotomy site and callus formation (black arrow). Mediolateral radiographs of control group subject immediately after TPLO (**b**) and after 8 weeks (**d**), showing a minimal degree of callus formation (white arrow).

**Figure 2 animals-13-02084-f002:**
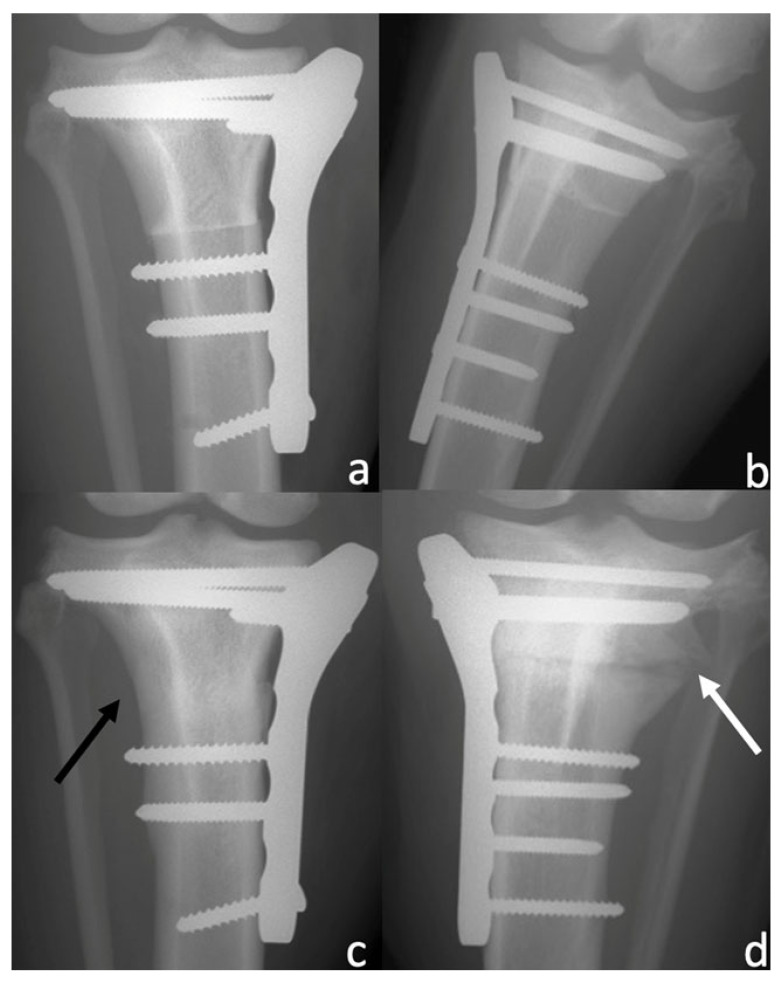
Caudocranial view of subjects treated with microfragmented adipose tissue (MFAT) immediately after tibial plateau levelling osteotomy (TPLO) (**a**) and after 8 weeks (**c**), demonstrating no osteotomy line visibility (black arrow). Caudocranial radiographs of control group subject immediately after TPLO (**b**) and after 8 weeks (**d**), demonstrating osteotomy line visibility (white arrow).

**Figure 3 animals-13-02084-f003:**
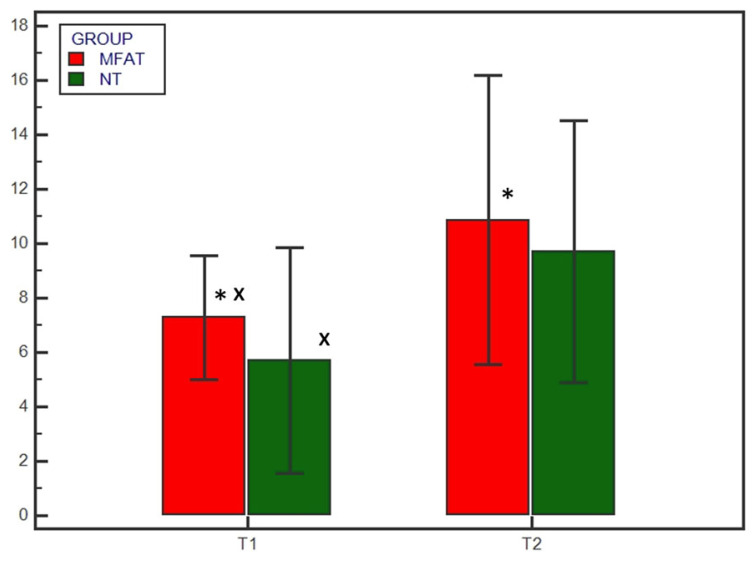
Radiographic healing based on the 12-point scoring system at T1 (4 weeks) and T2 (8 weeks) for the microfragmented adipose tissue (MFAT) group (red) and the control (NT) group (green). * indicates a significant difference between the groups (*p* < 0.05); x indicates a significant difference within the same group between T1 and T2 (*p* < 0.05).

**Figure 4 animals-13-02084-f004:**
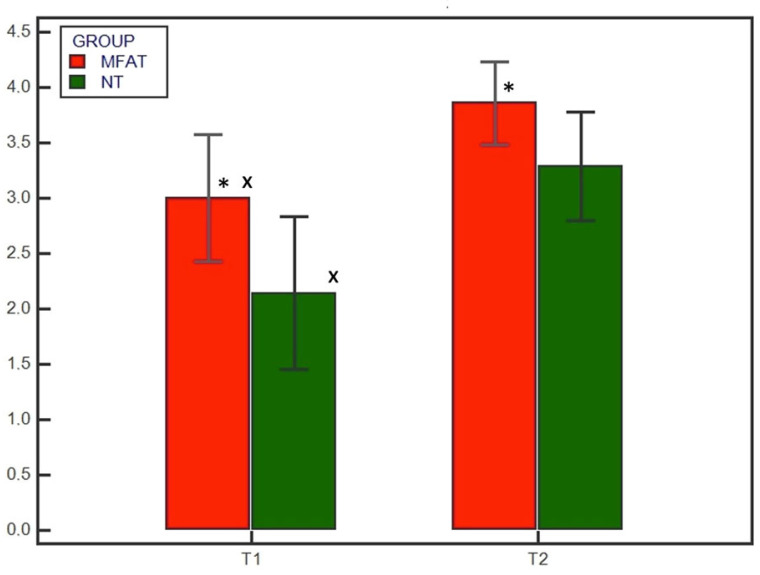
Radiographic healing based on the 5-point scoring system at T1 (4 weeks) and T2 (8 weeks) for the microfragmented adipose tissue (MFAT) group (red) and the control (NT) group (green). * indicates a significant difference between the groups (*p* < 0.05); x indicates a significant difference within the same group between T1 and T2 (*p* < 0.05).

**Figure 5 animals-13-02084-f005:**
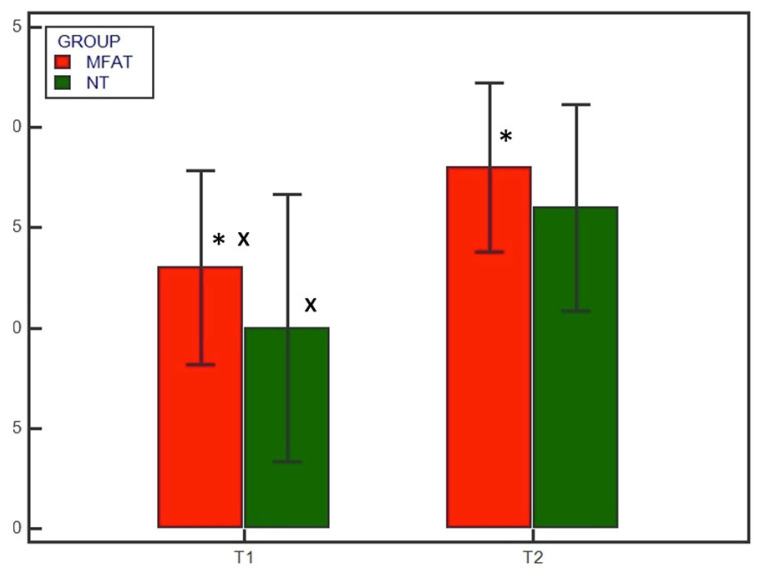
The degree of rounding of the distal step at the osteotomy site at T1 (4 weeks) and T2 (8 weeks) for the microfragmented adipose tissue (MFAT) group (red) and the control (NT) group (green). * indicates a significant difference between the groups (*p* < 0.05); x indicates a significant difference within the same group between T1 and T2 (*p* < 0.05).

**Figure 6 animals-13-02084-f006:**
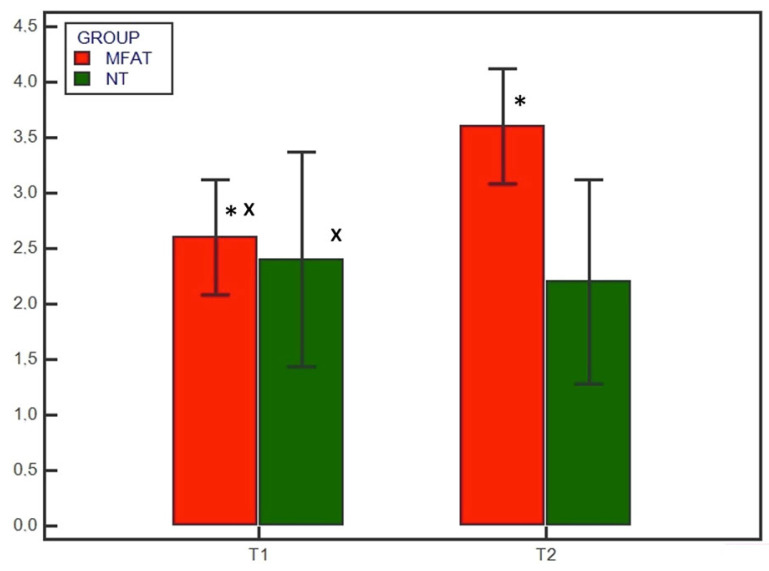
The subjective degree of callus formation at T1 (4 weeks) and T2 (8 weeks) for the microfragmented adipose tissue (MFAT) group (red) and the control (NT) group (green). * indicates a significant difference between the groups (*p* < 0.05); x indicates a significant difference within the same group between T1 and T2 (*p* < 0.05).

**Table 1 animals-13-02084-t001:** The 12-point scoring system with the specific parameters evaluated for bone healing [19,22].

12-Point Scoring System	
Cortical continuity (0–4)	0 = no cortical continuity
	1 = one continuous cortex
	2 = two continuous cortices
	3 = three continuous cortices
	4 = four continuous cortices
Osteotomy line visibility (0–2)	0 = osteotomy line visibility
	1 = osteotomy line barely visible
	2 = no osteotomy line visible
Subjective degree of callus formation or remodelling (0–4)	0 = none
	1 = minimal
	2 = moderate
	3 = remodelled
	4 = healed
Degree of rounding of the distal step at the osteotomy site (0–2)	0 = none
	1 = mild
	2 = significant

**Table 2 animals-13-02084-t002:** The 5-point scoring system with the percentage of bone healing associated with each score [22].

5-Point Scoring System	
0	No healing
1	1–25% healing
2	26–50% healing
3	51–75% healing
4	76–100% healing

## Data Availability

The clinical data used to support the findings of this study are included within the article.

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
