# Peer review of "Effects of Autologous Microfragmented Adipose Tissue on Healing of Tibial Plateau Levelling Osteotomies in Dogs: A Prospective Clinical Trial"

_animals, 2023, doi:10.3390/ani13132084_

Round 1

Reviewer 1 Report

First of all, I must express that the article under review deals with a very interesting subject and that has been little addressed in this specific technique for the craneal cruciate ligament rupture. In that way, the main doubt aboout this works and I would rather have seen some comments in the text is the manner in which the ossification of the tibial osteotomy is affected by the fact that stem cells have been introduced intra-articularly. 

In addition, perhaps it would have been interesting to consider the degree of chronicity of the lesion, as well as the degree of OA present. On the other hand, in my opinion it should have been compared with an objective biomechanical evaluation at that time, to see if it affects in any way the support of that limb, mainly by having a low sample number.   Regarding more specific things, the following things should be highlighted:   - Line 56: when yo say: `Strategies to treat these.....´It would be nice if you consider putting a bibliographical citation to postulate that sentence.
- Line 166: I would consider writing the bibliographical reference of the scale in the table as well. - Line 170: I would consider writing the bibliographical reference of the scale in the table as well. - In the Results: when you write about the `Enrolled patients (point 3.1)´: In my opinion, I would prefer to separate it into 2 paragraphs, the first being the one dealing with population variables, and the next one describing surgical variables. - Figure 1. I think I would have a better visualization if I separated into 2 frames, in which I would compare the mediolateral views for both groups on the one hand and the caudocranial views on the other hand. Also, pay attention to unify the use of lowercase or uppercase when naming them in the legend of the figure, with respect to what is seen in the photos. - Figuras 2, 3, 4 and 5: I would eliminate the superscript that is exposed above the graph, I understand that the figure is already described in the legend. - In the paragraph that begins on line 285 it seems to me that he is talking about the same thing as the previous paragraph. I don't know if I could find a way to join them together so that it doesn't look like a repetition.   Despite these small findings, I want to congratulate the authors for their great work.    

Author Response

Dear reviewer

Than you for the correction and suggestion.

The replies to comments are in the attached file and in the new version of manuscript (please see the attachment).

Reviewer 2 Report

Dear authors, this is an interesting pilot study on a relative small number of patients; there are, however, several procedural/methodical issues which require explanation and correction. I listed recommendations and suggestions within the text (in cursive print) using the "revision" tool in Word. Please find the file attached. Here, just a few comments on the more important issues which you will also find contextually in the manuscript: title: please adapt (blinded, randomised is a scientific standard and not the exception and does not need highlighting); line 148 to improve validity the control group, an injection of a similar volume of saline should have been done with the same 18 gauge needle); line 154 that means that in the study you enrolled retrospectively or prospectively only those cases with an “ideal” radiographic aspect of the osteotomy – this adds an important bias to the study – how many cases, and in which group, did you exclude because of insufficiently appearing compression ; line 160 was there no radiologist involved ? that the surgeon(s) who were involved in the procedures also evaluate the outcomes radiographically is a definite flaw; line 195 All subjects showed satisfying radiographic compression of tibial osteotomy immediately after surgery (T0) (of course they did because you excluded all others !!! see line 154); line 353 an osteotomy is NOT a traumatic fracture and should not be compared – or extrapolated in biologic bone healing studies. Hoping that you will be able to adapt and revise the manuscript accordingly because conceptually this study is of interest to specialist surgeons. 

made some corrections in the text but overall ok

Author Response

(The authors gave the same response as above.)

Round 2

Reviewer 2 Report

Thank you for the thorough revision; recommend publication as is.